



# Retrievals of $X_{CO_2}$, $X_{CH_4}$ and $X_{CO}$ from portable, near-infrared Fourier transform spectrometer solar observations in Antarctica.

David F. Pollard[1], Frank Hase[2], Mahesh Kumar Sha[3], Darko Dubravica[2], Carlos Alberti[2], and Dan Smale[1]

[1]National Institute of Water & Atmospheric Research Ltd (NIWA), Lauder, New Zealand
[2]Karlsruhe Institute of Technology, IMK-ASF, Karlsruhe, Germany
[3]Royal Belgian Institute for Space Aeronomy (BIRA-IASB), Brussels, Belgium

**Correspondence:** David F. Pollard (dave.pollard@niwa.co.nz)

**Abstract.** The Collaborative Carbon Column Observing Network (COCCON), uses low-resolution, portable EM27/SUN Fourier Transform Spectrometers (FTSs) to make retrievals of dry air mole fractions (DMFs, represented as $X_{gas}$) of $CO_2$, $CH_4$, $CO$ and $H_2O$ from near infrared solar absorption spectra. The COCCON has developed rapidly over recent years and complements the Total Carbon Column Observing Network (TCCON).

In this work we provide details of the first seasonal timeseries of near infrared $X_{CO_2}$, $X_{CH_4}$ and $X_{CO}$ retrievals from measurements made in Antarctica during the deployment of an EM27/SUN to the Arrival Heights laboratory on Ross Island (77.83° S, 166.66° E, 205 m AMSL) over the austral summer of 2019/20 under the auspices of the COCCON.

The DMFs of all three species were lower in Antarctica than at mid-latitude and for $X_{CO_2}$ and $X_{CO}$ the retrieved values were less variable. For $X_{CH_4}$ however, the variability was significantly greater and it was found that this was strongly correlated to 10     the proximity of the polar vortex.

In order to ensure the stability of the instrument and the traceability of the retrievals, side-by-side comparisons to the TCCON station at Lauder, New Zealand (45.04° S, 169.68° E, 370 m AMSL) and retrievals of the Instrument Line Shape (ILS) were made before and after the measurements in Antarctica. These indicate that over the course of the deployment the instrument stability was such that the change in retrieved $X_{CO_2}$ was well below 0.1%.

The value of this data for satellite validation is demonstrated by making comparisons with the Tropospheric Monitoring Instrument (TROPOMI) on the Sentinel-5 Precursor (S5P) satellite.

The data set is available from the COCCON Central Facility hosted by the ESA Atmospheric Validation Data Centre (EVDC) https://doi.org/10.48477/coccon.pf10.arrivalheights.R02 (Pollard, 2021).

## 1 Introduction

Precise, ground-based measurements of column averaged dry-air mole fractions of greenhouse gases, such as those produced by the Total Carbon Column Observing Network (TCCON, Wunch et al. (2011)), are essential for the validation of satellite measurements including those of the Greenhouse Gases Observing Satellite (GOSAT, Yokota et al. (2009)), the Orbiting



Carbon Observatory (OCO) 2 (Crisp et al., 2008) and 3 (Eldering et al., 2018) and the Tropospheric Monitoring Instrument (TROPOMI, Veefkind et al. (2012)).

While the TCCON have long been considered the gold standard in ground-based, near infrared, column averaged validation data, the size and cost of the high resolution Bruker IFS125HR instruments it is based on, combined with the supporting infrastructure required to operate them, have meant that only a limited number of instruments and sites have been established globally.

    The Bruker EM27/SUN is a portable low-resolution Fourier Transform Spectrometer (FTS) with a built-in solar tracker that

can measure near infrared, solar absorption spectra. From these spectra it is possible to retrieve column averaged dry air mole fractions (DMFs, represented as $X_{gas}$) of $CO_2$, $CH_4$, $CO$ and $H_2O$ with a precision and accuracy similar to or better than the TCCON (Gisi et al. (2012) and Hase et al. (2016)). A network based on the EM27/SUNs is being developed known as the Collaborative Carbon Column Observing Network (COCCON, Frey et al. (2019)).

    The comparatively low cost, portability and relative ease-of-use of these instruments has meant that they can be utilised in

greater numbers or to cover specific targets in locations where the more complex instruments cannot be deployed. For example, Velazco et al. (2019) used an EM27/SUN in a semiarid region of Australia to validate retrievals from GOSAT, Knapp et al. (2021) have conducted observations from a ship transiting the Pacific Ocean and Frey et al. (2021) have established a COCCON site in Namibia while Hase et al. (2015) and Dietrich et al. (2021) have used dense networks of these instruments to estimate the carbon fluxes of Berlin and Munich respectively. Tu et al. (2022) have gone on to demonstrate the utility of these instruments

to quantify fluxes at the facility level.

    In this work we present the data gathered during the deployment of an EM27/SUN to the Arrival Heights laboratory on Ross Island, Antarctica (77.83° S, 166.66° E, 205 m AMSL) over the austral summer of 2019/20. The retrieved time series of $X_{gases}$ is available from the COCCON Central Facility hosted by the ESA Atmospheric Validation Data Centre (EVDC) https://doi.org/10.48477/coccon.pf10.arrivalheights.R02 (Pollard, 2021). This dataset represents the first seasonal timeseries of

$X_{gases}$ retrieved from near infrared solar spectra in Antarctica and will provide a useful source of validation data at these high, southern latitudes which are not well covered by existing networks (The southernmost TCCON station is Lauder at 45°South).

    In the next section we will introduce the EM27/SUN instrument and briefly describe the retrieval scheme used to derive $X_{gases}$ from the measured solar spectra, as well as the other data sets we will compare the EM27/SUN results with. The following section will describe the measurements made at Arrival Heights, discuss the results and demonstrate the robustness

and utility of the data. Conclusions will be drawn in Sect. 4.

## 2   Instrumentation and Data Processing

In this section we will briefly describe the EM27/SUN instrument, followed by the retrieval scheme used to infer the column averaged dry air mole fractions of trace gases from the measured solar absorption spectra.





We will also provide a high level overview of the TCCON data from the Lauder site, which were compared to the EM27/SUN
retrievals before and after the period of deployment, and Sentinel 5 precursor data that were compared to the data collected at
Arrival Heights.

## 2.1  EM27/SUN Fourier Transform Spectrometer

The Bruker EM27/SUN is a portable, low-resolution Fourier transform spectrometer with a built-in solar tracker using a camera
trained on the detector aperture to provide active feedback. The instruments measure DC coupled interferograms with a spectral
resolution of $0.5\,\mathrm{cm^{-1}}$ using two Indium Gallium Arsenide (InGaAs) detectors at room temperature, one with a spectral range
of $5500–11000\,\mathrm{cm^{-1}}$ and a second, wavelength extended detector measuring in the $4000–5500\,\mathrm{cm^{-1}}$ range (Gisi et al. (2012)
and Hase et al. (2016)).

## 2.2  EM27/SUN Data Processing

The raw interferograms are processed and Xgas retrievals made using the PROFFAST software of the COCCON/PROCEEDS
framework which was developed on behalf of the European Space Agency (ESA) and is open-source and freely available. It has
previously been described by Sha et al. (2020) and will be summarised herein. The DC coupled, double sided interferograms
are processed into spectra by the PROFFAST-PREPOCESS code which includes a DC correction, phase correction and quality
checks of the resulting spectra. Separately, molecular absorption cross sections are calculated for each day by the PCXS module
based on the meteorological and trace gas priors generated using the TCCON method and for a representative surface pressure.
The trace gas retrievals are then computed by the INVERS module which uses a least squares fitting algorithm to scale the
prior profiles, with the option to adjust the surface pressure to that which was actually measured at the time of the observation.

A further correction is applied to the retrieved $X_{CO_2}$ values. This is to account for a compounded correction factor applied
by PROFFAST, which had been derived for an earlier version of the code. The earlier version included a bug which caused a
pointer offset of magnitude 1 in the spectrum handling. This bug was fixed in PROFFAST version 2020-08-10, and while it
was found that the correction factors applied to all other retrieved species remained valid, it was necessary to apply a further,
solar zenith angle dependent, correction to $X_{CO_2}$ of the form:

$$X_{CO_2corr} = X_{CO_2} \times \left( 1.0018 - 0.001 \times \left( \frac{SZA}{90} \right)^2 \right) \tag{1}$$

Where $X_{CO_2corr}$ and $X_{CO_2}$ are the corrected and uncorrected values and $SZA$ is the solar zenith angle in degrees. This
correction has been applied to the R02 version of the dataset described herein. A previous version (R01) was published before
the need to apply the correction was known. Further details of this correction and the background to it are available in "Technical
note on $X_{CO_2}$ bias in current PROFFAST distribution" (https://www.imk-asf.kit.edu/english/3225.php, last access: 27 January
2022).





## 2.3 TCCON

The Lauder TCCON station (45.04° S, 169.68° E, 370 m AMSL) is the southernmost in the network and has one of the longest
continuous records. The station has been previously described in Pollard et al. (2017) and Pollard et al. (2021). In this work
we have used the standard output from the GGG2014 data processing version (Wunch et al., 2015) which is publicly available
from the TCCON data archive (Pollard et al., 2019).

## 2.4 Sentinel 5 Precursor

ESA's Sentinel 5-precursor (S5P) satellite, which carries the TROPOMI instrument as its sole payload, is orbiting in a sun-
synchronous, low-earth polar orbit and has an observation swath of 2600 km wide across track resulting in daily global coverage
and a pixel size of 5.5 x 7 km for $CH_4$ and $CO$.

    The S5P operational $X_{CH_4}$ data are retrieved using the RemoteTeC-S5P algorithm (Hu et al., 2016), which produces re-
trievals of $X_{CH_4}$ only under cloud free conditions. In this work, we compared both the standard product and a bias corrected
version of the dataset. The details of the bias correction can be found in the algorithm theoretical baseline document (ATBD)
for S5P methane retrievals (Hasekamp et al., 2021) The S5P operational total column density of carbon monoxide (CO) are
retrieved using shortwave infrared carbon monoxide retrieval (SICOR) algorithm (Landgraf et al., 2016). The retrievals of $CO$
are performed simultaneously with interfering trace gases and effective cloud parameters, such as cloud height and optical
thickness. The offline (OFFL) operational data version 01.03.02 has been used in this paper.

## 3 Methods and Results

In this section we will first demonstrate the instrumental stability of both the EM27/SUN and the TCCON instrument at the
Lauder site through the Instrument Line Shape (ILS) retrieved for both.

    The experimental set ups used for the measurements made at Arrival Heights, and at Lauder before and after the deployment,
will be described in the subsequent two subsections along with discussion of the retrievals resulting from those measurements.

    Comparisons to the S5P retrievals will also be examined to demonstrate the application of this dataset to satellite validation
activities.

## 3.1 Instrument Stability

In order to monitor the stability of the alignment of an FTS, the ILS can be retrieved (Hase et al., 2013).

    For the Lauder TCCON instrument, regular (approximately monthly) measurements are made of an internal cell, containing
a calibrated quantity of HCl, illuminated by a lamp source. From these spectra, the ILS is retrieved using version 14.5 of the
LINEFIT software described in Hase et al. (2013).

    The EM27/SUN does not have the capability to measure HCl cells, but an ILS retrieval can be achieved by taking long
(4 m) path measurements of a lamp source and making use of water vapour adsorption as described in Frey et al. (2015) and

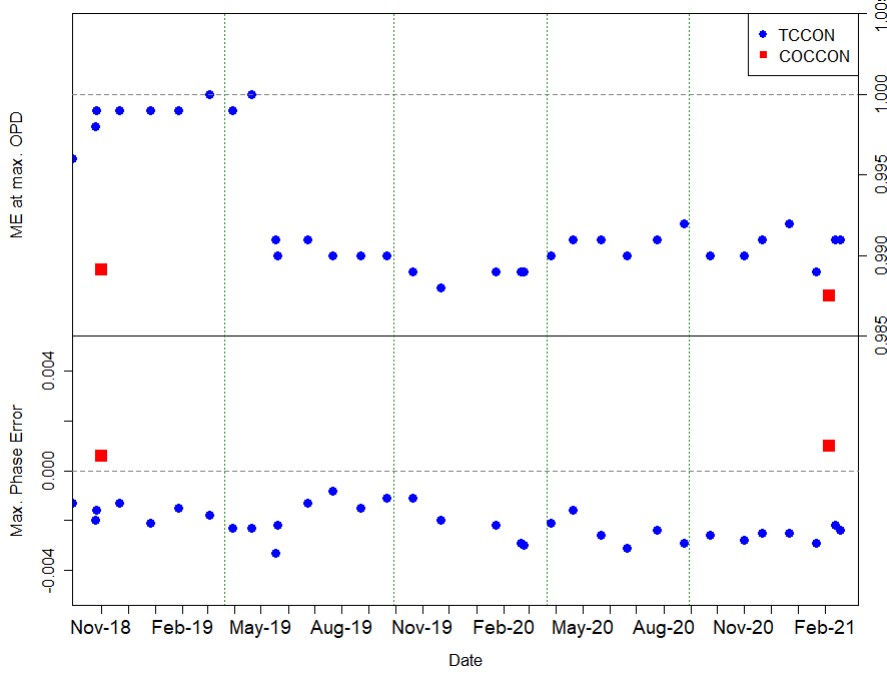

**Figure 1.** Time series of retrieved ILS parameters, modulation efficiency at maximum OPD (top panel) and maximum phase error (lower panel), for both instruments covering the period of this study. Vertical lines show the dates that the EM27/SUN was shipped from Karlsruhe to Lauder, Lauder to Arrival Heights, Arrival Heights to Lauder and Lauder to Karlsruhe respectively.

updated by Alberti et al. (2021). For the EM27/SUN instrument used in this study (serial number 53), ILS measurements were conducted at the Karlsruhe Institute of Technology before and after the instrument was shipped to New Zealand.

115     A time series of the results of these ILS retrievals, parameterised in terms of the modulation efficiency (ME) at maximum optical path difference (OPD) and the maximum value of the phase error, for both instruments are shown in fig. 1. A noticeable feature of fig. 1 is a step change in the ME at max. OPD for the TCCON instrument in May 2019. This is related to a change of the instrument's metrology laser on 28th May. Before this change, the mean ME at max. OPD was 0.9991 (with a standard deviation of 0.0006) after it was 0.9902 (0.0010). This one percent change in ME at max. OPD will not have a significant

120     impact on $X_{gas}$ retrievals as Hase et al. (2013) estimated that a 4% change would result in only a 0.035% error in $X_{CO_2}$. The TCCON maximum phase error remained virtually the same across this change at -0.0019 (0.0004) before and -0.0022 (0.0007) after. The EM27/SUN had an ME at max. OPD and max. phase error of 0.9891 and 0.0006 before leaving Karlsruhe and 0.9875, 0.0010 when it returned. We therefore conclude that both instruments maintained their alignment throughout the presented dataset.

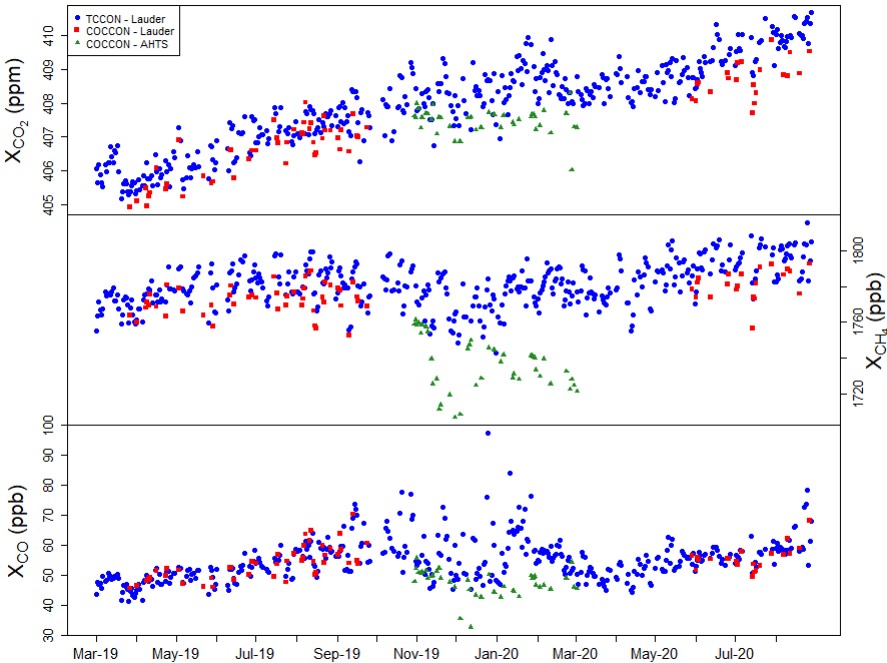

**Figure 2.** Time series of daily averaged $X_{CO_2}$ (top), $X_{CH_4}$ (middle) and $X_{CO}$ (bottom) retrieved form spectra measured using the Lauder
TCCON instrument (blue circles), the EM27/SUN whilst it was at Lauder (red squares) and when it was at the Arrival Heights laboratory
(green triangles)

## 3.2 EM27/SUN observations and retrievals at Arrival Heights

At Arrival Heights, the EM27/SUN was positioned on a bench inside the laboratory with a small amount of the beam from the
solar tracker (Robinson et al., 2020) used for a Bruker IFS125HR that is part of the Network for the Detection of Atmospheric
Composition Change (NDACC, De Mazière et al. (2018)) diverted into the EM27/SUN's solar tracker which was fixed in
position.

Measurements were taken on clear-sky days when a technician was present at the laboratory. Over the course of the 2019-20
summer season. Between 6[th] November 2019 and 9[th] March 2020 measurements were made on 41 days.

Under normal operations, ten interferogram scans over a period of about 70 secs are co-averaged into a single EM27/SUN
measurement. To halve the amount of data needing to be transferred, at Arrival Heights it was decided to co-average 20
interferograms. As the rate-of-change in solar zenith angle is relatively small at high latitudes, this change is not expected to
have any impact on retrievals.

The pressure data used in the Arrival Heights retrievals is taken from the weather station at nearby Scott Base (NIWA elec-
tronic weather station, EWS, Scott Base, ID:12740, 77.85° S, 166.76° E, 20 m AMSL) and corrected for the altitude difference
of 185 m between the two locations.





Figure 2 shows the time series of each of the Xgases retrieved from the EM27/SUN and the Lauder TCCON station before,
during and after the measurement campaign at Arrival Heights.

$X_{CO_2}$ measured at Arrival Heights is systematically lower than at Lauder. This is an expected result as high latitude southern
hemisphere $CO_2$ concentrations are less than at mid latitudes (Stephens et al., 2013). Also the seasonality that can be seen at
the lower latitude is not obvious in these measurements. However, over the period of the deployment, the growth rate of $X_{CO_2}$
values is also not seen at Arrival Heights and this is potentially because the seasonal draw down is of a similar magnitude and
negates it.

$X_{CH_4}$ shows considerable variability over the period of the deployment and a negative trend which contradicts the positive
trend seen at lower latitudes. To investigate this structure further we examined the isentropic modified potential vorticity
(MPV) (Lait, 1994) over Arrival Heights derived from the Modern Era Retrospective-Analysis for Research and Applications
reanalysis product (MERRA2) (Gelaro et al., 2017). Although not an absolute diagnostic of the position of the polar vortex
relative to Arrival Heights, the MPV value will be negatively correlated with the influence of the vortex (Smale et al., 2021).
The top panel of figure 3 shows the MPV at the 460K isentropic level (corresponding to the lower stratosphere) over the course
of the instrument campaign, while the lower panel shows the daily averaged $X_{CH_4}$ retrievals for the same period. There is clear
correlation between the two (r=0.82, 95% CI: 0.68-0.90). The correlation of $X_{CH_4}$ with MPV confirms a weak barrier effect
of the polar vortex (Choi et al., 2002).

The lower panel of figure 2 shows $X_{CO}$. The data collected at Arrival Heights is clearly measuring baseline $X_{CO}$ concentra-
tions whilst at lower latitudes, the Lauder TCCON data shows spikes caused by stratospheric transport of air masses affected
by biomass burning in the tropics.

### 3.3 Comparison to TCCON

Before and after the EM27/SUN was deployed to Arrival Heights, it was operated for a period at Lauder, alongside the TCCON
station there. Between 8th March and 22nd July 2019, the instrument was in a workshop where it could be moved outside on
fine days to make observations using the built-in solar tracker. An automatic scheduling application (Geddes et al., 2018) was
used to run the measurements and this was able to interrogate the output from the EM27/SUN's camtracker software and
pause measurements when cloud obscured the Sun. On 22nd July the EM27/SUN was relocated to the same laboratory as
the TCCON instrument with the built-in solar tracker parked and illuminated by a small amount (approximately 5%) of the
parallel solar beam from a solar tracker coupled to another Bruker 125HR. The difference in altitude between these locations
is 10 m and corrections have been applied to account for this in the surface pressure used for trace gas retrievals. This pressure
measurement is from the Vaisala PTB100A sensor that is part of the Lauder climate station.

After returning to New Zealand from Antarctica the instrument was returned to Lauder where it was installed under another,
permanently installed solar tracker and further measurements were collected from 5th June until 3rd September 2020 when it
was returned to KIT.

In total measurements were collected on 72 days alongside the Lauder TCCON station. To make a meaningful comparison
between the EM27/SUN and TCCON, we first average the retrievals from both instruments into ten-minute bins. The window



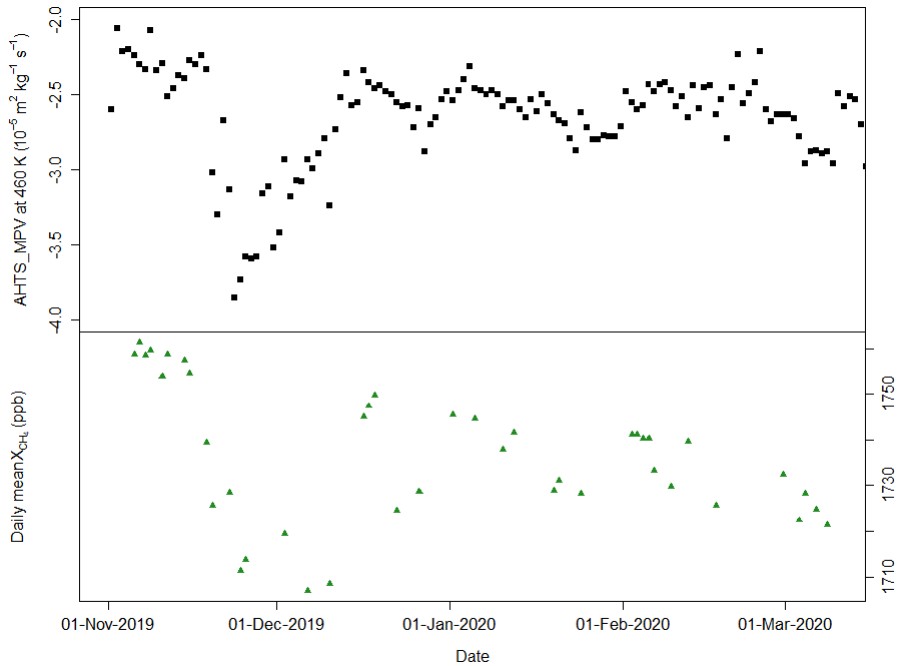

**Figure 3.** Daily MPV (AHTS_MPV, upper panel) and daily mean $X_{CH_4}$ retrieved during the period of the EM27/SUN deployment at Arrival Heights (lower panel)

of ten minutes was chosen to include sufficient measurements to reduce the effects of random uncertainties whilst not aliasing in slowly varying signals due to e.g. airmass dependence.

Figure 4 shows a timeseries of the differences between the 10 minute averages before and after the deployment for the three species, with box and whisker plots of these in the inset panel. A summary of the before and after statistics is also given in table 1. There is a small, but statistically significant difference for all three species between the before and after comparisons. As described previously, retrievals of the instrument line shape (ILS) for both instruments spanning the period in questions show no large drifts in the alignment of either. Therefore it is reasonable to expect that this is due to a seasonally dependent

bias between the two instruments. While it is difficult to draw conclusions from this limited time span plotted in fig. 4, it is not unreasonable to suggest this is the case. Such an effect was seen previously by Sha et al. (2020) when comparing low and high resolution instruments. This effect is likely caused by the averaging kernels of the low and high resolution instruments aliasing different errors in the common priors into the respective retrievals. In any case, the change in offset with respect to TCCON is 0.07% for $X_{CO_2}$ (0.16% for $X_{CH_4}$ and 1.72% for $X_{CO}$) which is below the 0.1% precision value generally accepted as the

requirement for carbon cycle studies and satellite validation (Wunch et al., 2015).


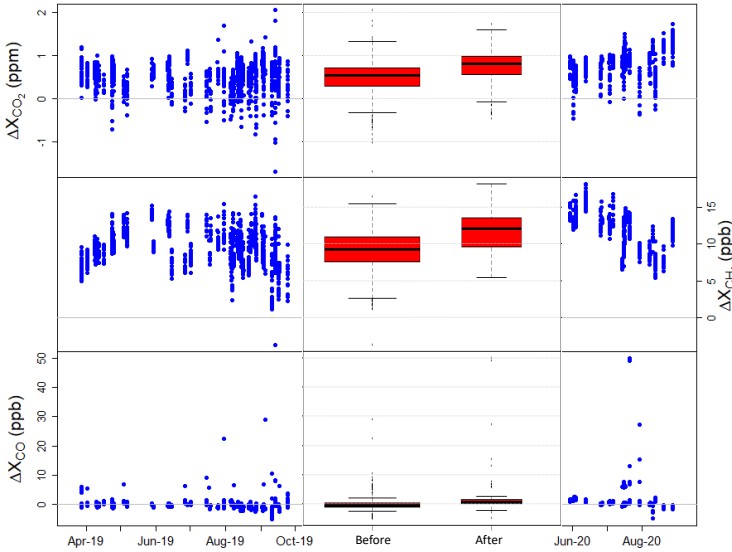

**Figure 4.** Time series of the difference between 10 minute averages of $X_{CO_2}$ (top), $X_{CH_4}$ (middle) and $X_{CO}$ (bottom) retrieved by the TCCON and EM27/SUN instruments at Lauder before and after the deployment to Arrival Heights. The inset panel shows box and whisker plots of these differences

**Table 1.** TCCON - COCCON comparison statistics (median and standard deviation) before and after the EM27/SUN deployment to Arrival Heights

| Species | Before | | After | |
|---|---|---|---|---|
| | med | sd | med | sd |
| $X_{CO_2}$ (ppm) | 0.535 | 0.342 | 0.805 | 0.370 |
| $X_{CH_4}$ (ppb) | 9.25 | 2.57 | 12.09 | 2.70 |
| $X_{CO}$ (ppb) | -0.485 | 1.789 | 0.469 | 3.930 |

### 3.4 Comparison to S5P

The spatial and temporal coincidence criteria used to select S5P measurements corresponding to EM27/SUN observations are the same as Sha et al. (2021), i.e. within a radius of 100 km (50 km) around the site for methane (carbon monoxide) validation and with a maximal time difference of 1 h for EM27/SUN observations around the S5P overpass time. Because of the high

latitude of the Arrival Heights laboratory and the inclination of the S5P orbit, this often results in coincidences on more than one S5P orbit for a particular day's EM27/SUN observing period.





The average S5P pixel values are compared to EM27/SUN retrievals which have had the a priori alignment applied to compensate or correct for its contribution to the smoothing equation (Rodgers and Connor, 2003). The co-located pairs are selected only if a minimum of five S5P pixels were found in applying the co-incidence criteria.

Figure 5 shows the comparison between the ground based EM27/SUN and the S5P retrievals for bias corrected $X_{CH_4}$ and standard $X_{CO}$ (left and right plots respectively) with all values retrieved by each instrument (pale symbols) as well as co-located overpass means (bold) (upper panels) and the relative differences (lower panels).

     The bias corrected S5P $X_{CH_4}$ product show a bias of 2.14%, which exceeds the S5P bias requirement (1.5%). The bias of the standard S5P $X_{CH_4}$ product without the albedo dependent correction is 1.05%. This shows that the S5P $X_{CH_4}$ products

at this location is strongly dependent on the surface albedo and the respective correction applied to the product. The standard deviation of the relative bias, which is a measure of the random error, is below 0.3% for both standard and bias-corrected S5P $X_{CH_4}$ products, which is well below the requirement of 1%. This high bias at Arrival Heights was also seen by Sha et al. (2021) when they performed a comparison with the NDACC station there. This was attributed to the highly variable topography in the region. Similar effects have also been noted at high northern latitudes (Lambert et al., 2021, p. 129), again attributed to

variability on surface albedo and topography. These results further highlight the value of having reliable, high quality, ground-based measurements which are independent of surface effects to validate satellite retrievals, particularly in challenging regions such as these.

     For S5P $X_{CO}$, the mean bias is 3.77%, which is well within the S5P bias requirement (15%). The standard deviation of the relative bias is 4.73%, which is well below the requirement of 10%. Applying a cone co-location criterion to the S5P data

following the ground-based EM27/SUN line of sight, as recommended in Sha et al. (2021) for high latitude sites, we find a mean bias of 5.89% compared to the EM27/SUN. The bias with respect to the EM27/SUN is less than the bias of 11.99% that Sha et al. (2021) found when comparing to the NDACC data using a common a priori for a period of about three years. The discrepancy is mostly because of the different time periods between the NDACC and EM27/SUN comparisons, a priori difference or spectroscopy limitations. The S5P bias shows a seasonality with respect to NDACC data with high values in

September and October, a period when the EM27/SUN did not measure. Furthermore, a bias change of about +3.7% has been reported for the S5P validation with NDACC when comparing directly vs using the S5P a priori as the common prior, whereas this change is about -0.26% for EM27/SUN comparisons.

## 4    Conclusions

We have described the first seasonal timeseries of $X_{CO_2}$, $X_{CH_4}$ and $X_{CO}$ retrieved from near infrared solar spectra in Antarc-

tica gathered during the deployment of an EM27/SUN to the Arrival Heights laboratory on Ross Island, Antarctica over the austral summer of 2019/20 under the auspices of the COCCON.

     Through monitoring of the instrument ILS, and by comparing to the Lauder TCCON station before and after the deployment we have demonstrated that the precision of the $X_{CO_2}$ retrievals was 0.07% (0.16% for $X_{CH_4}$ and 1.72% for $X_{CO}$) over the duration of the measurement campaign, and that the instrument was unaffected by being shipped to and from Antarctica.

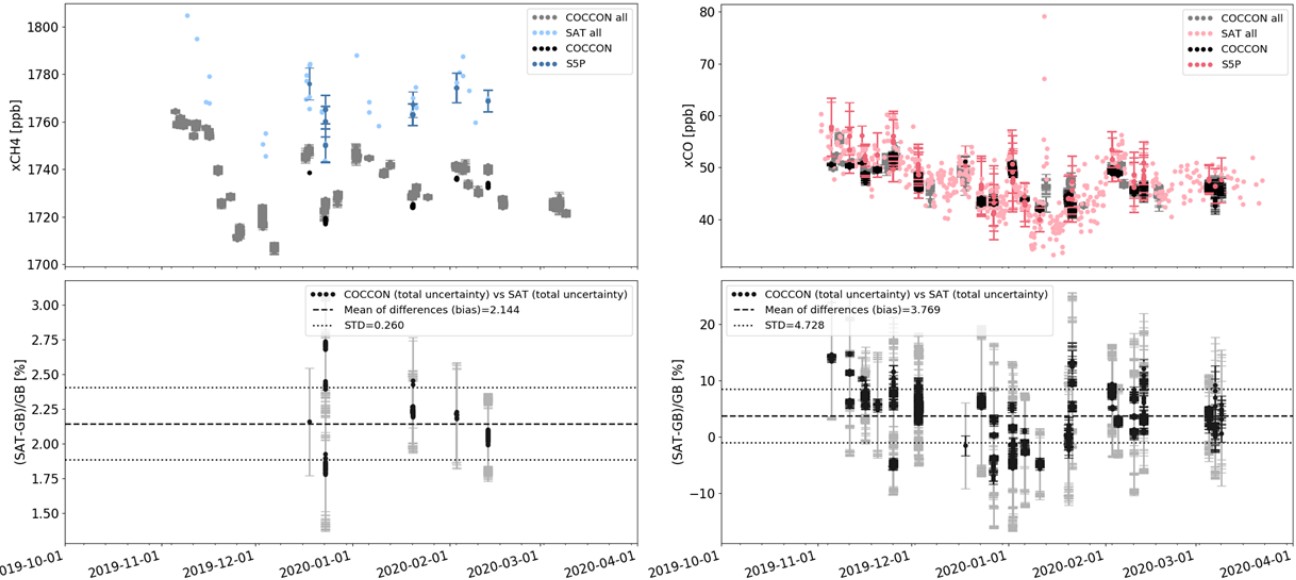

**Figure 5.** Comparison with Sentinel 5 Precursor retrievals of $X_{CH_4}$ (left column) and $X_{CO}$ (right column) showing all retrievals and overpass means (top panels) and the relative difference (lower panels)

The retrieved column averaged abundances of all three measured species ($X_{CO_2}$, $X_{CH_4}$ and $X_{CO}$) were lower in Antarctica than at the mid-latitude Lauder TCCON station as expected. However, the range of $X_{CH_4}$ values observed at Arrival Heights was larger than at Lauder and are well correlated with the proximity of the polar vortex edge.

When comparing the EM27/SUN retrievals at Arrival Heights to S5P it was found that the S5P, bias corrected $X_{CH_4}$ product had a mean difference of 2.14% which exceeded the mission bias requirement of 1.5%. However the product without the albedo

dependent bias correction only differed by 1.05%, suggesting that the albedo dependent bias correction is not valid under these surface conditions. This finding is consistent with previous studies and highlights the value of these high quality, ground-based measurements which are independent of surface effects to validate satellite retrievals. For S5P $X_{CO}$, the mean bias is 3.77%.

It is expected that further deployments of EM27/SUN instruments to Arrival Heights will be undertaken in the future and the data added to the COCCON archive.



## 5 Data availability

The EM27/SUN Arrival Heights data set is available from the COCCON Central Facility hosted by the ESA Atmospheric
Validation Data Centre (EVDC) https://doi.org/10.48477/coccon.pf10.arrivalheights.R02 (Pollard, 2021). The Lauder TCCON
data can be accessed at tccondata. https://doi.org/10.14291/tccon.ggg2014.lauder03.R0 (Pollard et al., 2019). The public S5P
CO data can be accessed via https://doi.org/10.5270/S5P-1hkp7rp (Copernicus Sentinel-5P, 2018) and the public S5P CH4
data can be accessed via https://doi.org/10.5270/S5P-3p6lnwd (Copernicus Sentinel-5P, 2019). Other data sets are available
from the authors on request.

*Author contributions.* DP operated the EM27/SUN at Lauder,installed it at Arrival Heights, conducted the data processing and analysis
and wrote this manuscript. MS performed the comparison with S5P, FH supplied the EM27/SUN and developed the PROFFAST code. DS
provided expertise on instrument deployment at Arrival Heights, derivation of the MPV timeseries and assisted in manuscript preparation.

*Competing interests.* The authors declare that they have no competing interests.

*Acknowledgements.* We would like to thank Antarctica New Zealand for providing logistical support for the measurements at Arrival Heights
and Hue Tran, Jamie McGaw and Mark Murphy for carrying out the measurements. The EM27 and TCCON measurements at Lauder and
Arrival Heights are core-funded by NIWA through New Zealand's Ministry of Business, Innovation and Employment Strategic Science
Investment Fund.



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
