# Peer review of "Retrievals of $X_{CO_2}$ , $X_{CH_4}$ and $X_{CO}$ from portable, near-infrared Fourier transform spectrometer solar observations in Antarctica."

_Earth System Science Data, 2022_

## Referee Comment (RC1)

Retrievals of $X$CO$_2$, $X$CH$_4$, and $X$CO from portable, near-infrared Fourier transform spectrometer solar observations in Antarctica.

Pollard et al. studied the data collected using portable Fourier transform spectrometer (Bruker EM27/SUN) at the site Arrival Heights laboratory, Antarctica. ILS was compared for both the portable FTS and TCCON instrument to study the instrument stability. TCCON data from the site , Lauder was compared to the FTS retrievals. Comparison of $X$CH$_4$ and $X$CO with TROPOMI was also studied by the authors.

**Minor comments**

1. In abstract section the data set reference lines no 17-28 is not necessary. It can be mentioned in the data section in the manuscript.
2. The latitude and longitude at Arrival heights and at TCCON station Lauder can be removed from the abstract section and can be added appropriately in the Introduction section.
3. In introduction section, Line 23, please club OCO-2 and OCO-3 and keep references later.
4. Line 25 ---While the TCCON have long been considered the gold standard…. Please check the sentence.  It may be  global standard….
5. Line 48- please check the sentence. It is not clear.
6. The objective of the study can be clearly written in the introduction section like comparison with the  TCCON station and with TROPOMI satellite data.
7. Please check the references format as per the journal style.
8. Page 7 sentence 144 is not clear. Please check the sentence and modify as required.
9. Section 2.2 sentence 79 is not clear. Please check and correct the sentence.
10. Please check fonts for all the figures, x-axis , y-axis, legend. Please follow journal guidelines for figures and tables.
11. Please check for $X$ ( dry air mole fraction ) which should be italic throughout the manuscript.

**Major comments**

1. Authors can make comparative study of CO$_2$ data with portable FTS with satellite data such as OCO-2 or any other satellite data available during the common period of measurements.
2. Figure 5,  please abbreviate the term SAT in the legend. This is not clear……
3. Please check for English grammar throughout the manuscript.

---

## Author Comment (AC1)

**Maps**

[Figure]

**Maps**

[Figure]

**Maps**

[Figure]

**Maps**

[Figure]

[Figure]

**Maps**

---

## Author Response (AR1)

We thank the reviewers for taking the time to review our manuscript and for their constructive and thought-provoking comments.

Below we have included the full text of their reviews as indented text, interspersed with our responses addressing their specific comments as non-indented text and changes to the manuscript in *italicised* font.

Anonymous Referee #1

> Pollard et al. studied the data collected using portable Fourier transform spectrometer (Bruker EM27/SUN) at the site Arrival Heights laboratory, Antarctica. ILS was compared for both the portable FTS and TCCON instrument to study the instrument stability. TCCON data from the site, Lauder was compared to the FTS retrievals. Comparison of XCH4 and XCO with TROPOMI was also studied by the authors.

> **Minor comments**

> 1. In abstract section the data set reference lines no 17-28 is not necessary. It can be mentioned in the data section in the manuscript.

This is a manuscript composition requirement for ESSD as detailed in item #2 at https://www.earth-system-science-data.net/submission.html#manuscriptcomposition.

> 2. The latitude and longitude at Arrival heights and at TCCON station Lauder can be removed from the abstract section and can be added appropriately in the Introduction section.

The coordinates have been removed from the abstract and maintained at the first mention of each location in the main text.

> 3. In introduction section, Line 23, please club OCO-2 and OCO-3 and keep references later.

We have changed the text to read: "*the Orbiting Carbon Observatory (OCO) 2 and 3 (Crisp et al. (2008) and Eldering et al. (2018))*"

> 4. Line 25 ---While the TCCON have long been considered the gold standard…. Please check the sentence. It may be global standard….

"gold standard" is the intended expression.

> 5. Line 48- please check the sentence. It is not clear.

We have changed this sentence to the following: "*In the next section we will introduce the EM27/SUN instrument, briefly describe the retrieval scheme used to derive Xgases from the measured solar spectra, and describe the other data sets we will compare the EM27/SUN results with.*"

> 6. The objective of the study can be clearly written in the introduction section like comparison with the TCCON station and with TROPOMI satellite data.

The objective of this manuscript is to describe and document the presented dataset, this is described in the penultimate paragraph of the introduction. The comparison to TCCON is included to

demonstrate the robustness and calibration of the dataset, whilst the TROPOMI comparison serves as an example of the usefulness of the data.

7. Please check the references format as per the journal style.

All references are stored in a bibtex bibliography and are automatically rendered to the journal style during compilation of the LaTex source.

8. Page 7 sentence 144 is not clear. Please check the sentence and modify as required.

The sentence has been modified as follows: "*However, over the period of the deployment, the growth rate of XCO2 is also not seen at Arrival Heights. This is potentially because the seasonal draw down is of a similar magnitude and obscures the growth over this short period.*"

9. Section 2.2 sentence 79 is not clear. Please check and correct the sentence.

Modified to: "*This correction has been applied to the R02 version of the dataset described in this work. A previous dataset version (R01) was published before the need to apply the correction was known.*"

10. Please check fonts for all the figures, x-axis , y-axis, legend. Please follow journal guidelines for figures and tables.

This has been done.

11. Please check for X ( dry air mole fraction ) which should be italic throughout the manuscript.

Errant, un-italicised *X*'s have been corrected at lines 64 and 139.

**Major comments**

1. Authors can make comparative study of CO2 data with portable FTS with satellite data such as OCO-2 or any other satellite data available during the common period of measurements.

There is only limited level-3 data from OCO-2 available at the latitudes relevant to this study (see the attached, supplementary material, which maps monthly average OCO-2 L3 data). Therefore, it is not possible to make any statistically meaningful comparisons to this, or data from other satellite instruments for the same reasons. This is partly due to the difficulty of making robust retrievals under these conditions but also, the lack of validation sources in these areas, which the dataset presented in this work will be able to address. We have included a comparison with TROPOMI retrievals, of which there are enough to make statistically valid comparisons, to demonstrate a valid use case for our data. However, the use of our dataset to modify the L3 retrieval characteristics of an existing satellite data product in order to increase the throughput of high-latitude observations goes well beyond the scope of this data description manuscript.

2. Figure 5, please abbreviate the term SAT in the legend. This is not clear……

The term "SAT" has been replaced with "S5P" in the legend of Fig. 5

3. Please check for English grammar throughout the manuscript.

This has been done.

Anonymous Referee #2

The paper "Retrievals of XCO2, XCH4 and XCO from portable, near-infrared Fourier transform spectrometer solar observations in Antarctica" by Pollard et al., ESSD

The paper described the EM27/SUN COCCON measurements carried out in Antarctica during a summer season in 2019/20. The EM27/SUN XCO2, XCH4, and XCO measurements were compared to TCCON measurements at Lauder and TROPOMI measurements at Antarctica. Overall, the paper is concise and easy to understand. However, this manuscript does not include too many new findings and discriminated features.

Major comments:

The significance of this dataset is not so clear. The data covers a summer season in Antarctica. The measurements are valuable in the polar region. However, the dataset is too short for climate change research. For satellite validation? The authors show the comparisons with TROPOMI XCH4 and XCO. And then? I expected to get more solid conclusions from their experiment.

We have presented the first season of data in what we hope will become a longer term timeseries as further instrument deployments are undertaken. As the reviewer states, these measurements are valuable in the polar region where there are no other permanent facilities making similar observations. We have gone on to demonstrate a use-case for the data by making comparisons to the level 2 TROPOMI data. We feel that we have adequately described and demonstrated the quality of the dataset consistent with the Aims and Scope of ESSD (https://www.earth-system-science-data.net/about/aims_and_scope.html), specifically: "Articles in the data section may pertain to the planning, instrumentation, and execution of experiments or collection of data. Any interpretation of data is outside the scope of regular articles."

The uncertainty and precision of the EM27/SUN measurements are not clear to me, either. The authors have compared the EM27/SUN to the gold-standard TCCON measurements at Lauder. The differences in XCO2, XCH4, and XCO do exist. So, can we trust the EM27/SUN measurements in Antarctica? As the TCCON version has been updated to GGG2020, the authors should use the latest version instead of GGG2014. As the authors said that the different averaging kernels may cause their difference. Sha (2020) used the truncated spectra to get an HR_LR product to reduce the smoothing error when comparing two datasets. Such kind of experiment can be done here too.

The precision of the EM27/SUN measurements is described at lines 184 and 223 of the initial submission version of the manuscript, while an indication of the accuracy is given in Table 1.

The question of whether the data can be trusted can easily be answered by comparing the stability of the EM27/SUN retrievals with respect to the TCCON with the accuracy requirements of the application the data are to be used for. To anticipate the accuracy requirement of an unspecified application would be beyond the scope of this work.

At the time the analysis for this manuscript was undertaken, GGG2014 was the current, publicly available, version of the TCCON data.

Minor comments:

There are many places with bad punctuation marks. Please check.

This has been done.

dry air mole fractions (DMFs ), add column-averaged dry air …

This change has been made at line 2 in the abstract.

Page 2 line 31. 'with a precision and accuracy similar to or better'. The authors said TCCON is the gold standard, so how EM27/SUN can be better than the gold standard?

Our statement that the precision + accuracy of the COCCON (EM27/SUN FTS) is equal or even better than TCCON (IFS125HR FTS) was meant as a short summary of a more complex performance metric when comparing TCCON and COCCON: Due to the lower resolution of the EM27/SUN in comparison to TCCON, the noise level on the primary target gases XCO2 and XCH4 (despite the lower interferometric etendue) is significantly lower than achieved by TCCON. In addition to that, the cadence of the measurements is higher. So, with respect to precision and amount of data per time interval, COCCON XCO2 and XCH4 products are superior. However, a well-aligned IFS125HR spectrometer is expected to achieve near-nominal instrumental characteristics. In the absence of technical errors, this guarantees that XCO2 data collected by a TCCON station agree on the ~0.05% level with the results that would be achieved by an imaginary collocated ideal TCCON spectrometer. The EM27/SUN shows a higher scatter of instrument-specific characteristics. This requires performing a characterisation of each EM27/SUN spectrometer by using side-by-side comparisons with a collocated TCCON station before use (Alberti et al. 2022). Without such calibration for achieving the connection to a TCCON reference spectrometer, there might be an instrument-specific calibration bias in the order of 0.1% for XCO2. Due to this fact, the calibration of TCCON is the gold standard and COCCON measurements are tied to the TCCON reference by the aforementioned determination of instrument-specific calibration factors for each target gas. Luckily, the deduced instrumental characteristics generally are stable on timescales of several years (Alberti et al. 2022), which makes the use of a previously calibrated COCCON spectrometer for verifying explicitly the expected level of consistency between different TCCON sites a sensible exercise. Indeed, unexpected biases of TCCON stations violating the targets of TCCON's internal network consistency have been uncovered by performing side-by-side comparisons of TCCON and COCCON spectrometers, e.g., a problem with detector nonlinearity at the TCCON site Sodankyla was discovered using this approach (Sha et al. 2020).

Page 4 line 91. Km -> km^2

The linear dimensions of the pixel are being described; therefore, km is appropriate. If the surface area were being enumerated, then $km^2$ would be appropriate.

Page 8 line 185. The authors address that the change in offset for XCO2 is not significant. How about XCH4 and XCO?

Whether or not the accuracy is significant depends on the application the data is to be used for. Because there is a significant body of work using measurements of CO2 to validate climate models (Rayner and O'Brien 2001; Olsen and Randerson 2004) the accuracy requirements that have been determined to achieve this have been adopted as an "industry-standard" for CO2 measurements. Such requirements have not been as rigorously determined for other species.

Fig4, Table1, and Fig5. Some use the absolute unit (ppm/b), and some use the relative unit (%). It is better to use a constant rule.

Throughout the manuscript we have tried to consistently use ppm for the absolute magnitude of XCO2 and ppb for XCH4 and XCO as these units give the most natural arrangement of significant figures. To make it more convenient for the reader we have used percentages to highlight differences. In the original submission this was not applied consistently and so we have changed the axes on Fig. 4 and the units of Tab. 1

References

Alberti, C., Hase, F., Frey, M., Dubravica, D., Blumenstock, T., Dehn, A., Castracane, P., Surawicz, G., Harig, R., Baier, B.C., Bès, C., Bi, J., Boesch, H., Butz, A., Cai, Z., Chen, J., Crowell, S.M., Deutscher, N.M., Ene, D., Franklin, J.E., García, O., Griffith, D., Grouiez, B., Grutter, M., Hamdouni, A., Houweling, S., Humpage, N., Jacobs, N., Jeong, S., Joly, L., Jones, N.B., Jouglet, D., Kivi, R., Kleinschek, R., Lopez, M., Medeiros, D.J., Morino, I., Mostafavipak, N., Müller, A., Ohyama, H., Palmer, P.I., Pathakoti, M., Pollard, D.F., Raffalski, U., Ramonet, M., Ramsay, R., Sha, M.K., Shiomi, K., Simpson, W., Stremme, W., Sun, Y., Tanimoto, H., Té, Y., Tsidu, G.M., Velazco, V.A., Vogel, F., Watanabe, M., Wei, C., Wunch, D., Yamasoe, M., Zhang, L., Orphal, J. (2022) Improved calibration procedures for the EM27/SUN spectrometers of the COllaborative Carbon Column Observing Network (COCCON). *Atmos. Meas. Tech.*, 15(8): 2433-2463. 10.5194/amt-15-2433-2022

Olsen, S.C., Randerson, J.T. (2004) Differences between surface and column atmospheric CO2 and implications for carbon cycle research. *Journal of Geophysical Research: Atmospheres*, 109(D2): n/a-n/a. 10.1029/2003JD003968

Rayner, P.J., O'Brien, D.M. (2001) The utility of remotely sensed CO2 concentration data in surface source inversions. *Geophysical Research Letters*, 28(1): 175-178. 10.1029/2000GL011912

Sha, M.K., De Mazière, M., Notholt, J., Blumenstock, T., Chen, H., Dehn, A., Griffith, D.W.T., Hase, F., Heikkinen, P., Hermans, C., Hoffmann, A., Huebner, M., Jones, N., Kivi, R., Langerock, B., Petri, C., Scolas, F., Tu, Q., Weidmann, D. (2020) Intercomparison of low- and high-resolution infrared spectrometers for ground-based solar remote sensing measurements of total column concentrations of CO2, CH4, and CO. *Atmos. Meas. Tech.*, 13(9): 4791-4839. 10.5194/amt-13-4791-2020